Morphological characterization of virus-like particles in coral reef sponges

Pascelli Cecília ceciliapascelli@gmail.com 1 2 3
Laffy Patrick W. 1
Kupresanin Marija 4
Ravasi Timothy 4
Webster Nicole S. 1 3 5
1 Australian Institute of Marine Science , Townsville , Queensland , Australia
2 James Cook University , Townsville , Queensland , Australia
3 AIMS@JCU, Australian Institute of Marine Science and James Cook University , Townsville , Queensland , Australia
4 KAUST Environmental Epigenetic Program (KEEP), Division of Biological and Environmental Sciences & Engineering, King Abdullah University of Science and Technology , Thuwal , Kingdom of Saudi Arabia
5 Australian Centre for Ecogenomics, University of Queensland , Brisbane , Queensland , Australia
Breitbart Mya
Electronic publication date: 2018 Oct 17
Publication date: 2018
Volume: 6
Electronic Location ID: e5625
Received 2018 Jun 28; Accepted 2018 Aug 22
Copyright: ©2018 Pascelli et al.
Copyright year: 2018
Copyright holder: Pascelli et al.
License: This is an open access article distributed under the terms of the Creative Commons Attribution License, which permits unrestricted use, distribution, reproduction and adaptation in any medium and for any purpose provided that it is properly attributed. For attribution, the original author(s), title, publication source (PeerJ) and either DOI or URL of the article must be cited.
License URL: https://creativecommons.org/licenses/by/4.0/

Keywords: Transmission Electron Microscopy, Virus, TEM, VLP, Marine sponges, Great Barrier Reef, GBR, Red Sea

Funding: Australian Research Council Future Fellowship FT120100480 The CAPES Foundation and Science Nicole S. Webster was funded through an Australian Research Council Future Fellowship FT120100480 and Cecília Pascelli was funded through a PhD Scholarship from The CAPES Foundation and Science without Borders. The funders had no role in study design, data collection and analysis, decision to publish, or preparation of the manuscript.

==============================
Marine sponges host complex microbial consortia that vary in their abundance, diversity and stability amongst host species. While our understanding of sponge-microbe interactions has dramatically increased over the past decade, little is known about how sponges and their microbial symbionts interact with viruses, the most abundant entities in the ocean. In this study, we employed three transmission electron microscopy (TEM) preparation methods to provide the first comprehensive morphological assessment of sponge-associated viruses. The combined approaches revealed 50 different morphologies of viral-like particles (VLPs) represented across the different sponge species. VLPs were visualized within sponge cells, within the sponge extracellular mesohyl matrix, on the sponge ectoderm and within sponge-associated microbes. Non-enveloped, non-tailed icosahedral VLPs were the most commonly observed morphotypes, although tailed bacteriophage, brick-shaped, geminate and filamentous VLPs were also detected. Visualization of sponge-associated viruses using TEM has confirmed that sponges harbor not only diverse communities of microorganisms but also diverse communities of viruses.

Introduction

Sponges are abundant and ecologically important members of marine benthic communities (Van Soest et al., 2012). Most sponges are suspension filter feeders (Thomassen & Riisgård, 1995), with complex aquiferous systems capable of manipulating the seawater composition at both macro and micro scales (Vacelet & Boury-Esnault, 1995; Patterson et al., 1997; De Goeij et al., 2013). A unidirectional (ostia-chamber-atrium-oscula) water flow driven by flagellated choanocyte cells is responsible for capturing and retaining small eukaryotes, prokaryotic cells and viral particles (Hadas et al., 2006). Sponge filtration of large quantities of seawater represents an important nutrient link between the pelagic and benthic environments (Pile & Young, 2006), especially in oligotrophic ecosystems such as coral reefs (De Goeij et al., 2013).

Sponges form intimate partnerships with diverse microbial consortia, and these relationships range from mutualism to commensalism to parasitism (Webster & Taylor, 2012; Thomas et al., 2016). The sponge microbiome is often highly conserved across individuals of the same sponge species but varies considerably across species (Thomas et al., 2016). It is because of these functionally important symbiotic partnerships that sponges are considered a typical example of a marine ‘holobiont’, an organism comprised of various ‘bionts’, living in symbiogenesis (Margulis & Fester, 1991; Webster & Thomas, 2016). However, while the symbiotic association between sponges and their bacterial/archaeal symbionts has been extensively studied (Schmitt et al., 2012; Thomas et al., 2016; Pita et al., 2018), the role of viruses in the sponge holobiont remain largely unknown, despite TEM images from the 1970s alluding to viral-infected sponge cells (Vacelet & Gallissian, 1978), a demonstration of phage infection in a sponge-associated bacterium ( Lohr, Chen & Hill, 2005), and a few recent metagenomic studies providing insights into sponge virus diversity and function (Butina et al., 2015; Laffy et al., 2016; Laffy et al., 2018).

Viruses are the most abundant biological agents in marine ecosystems, with about 1010 viruses per liter of surface seawater and 1010 per gram dry weight of marine sediment (Suttle, 2007; Danovaro et al., 2011). Importantly, viruses have the ability to regulate the prokaryotic and eukaryotic populations responsible for maintaining metabolic cycling in complex ecosystems such as coral reefs (Seymour et al., 2005; Thurber & Correa, 2011; Mojica & Brussaard, 2014). Viruses modulate microbial-driven processes through mortality, horizontal gene transfer and metabolic reprogramming by viral-encoded auxiliary metabolic genes (AMGs) (Bergh et al., 1989; Rohwer & Thurber, 2009; Danovaro et al., 2011; Hurwitz et al., 2014; Breitbart et al., 2018). Recent years have seen an increased focus on the diversity and function of viruses associated with reef invertebrates including sea anemones (Wilson & Chapman, 2001); starfish (Hewson et al., 2018); scleractinian corals and their associated microbial communities (Patten, Harrison & Mitchell, 2008; Weynberg et al., 2014; Weynberg et al., 2017a; Laffy et al., 2018). However, while viruses have been described as essential components of coral reef ecosystems, capable of controlling microbial community dynamics, playing a role in coral bleaching/disease, and mediating reef biogeochemical cycling (Thurber et al., 2017), there is a paucity of research exploring viruses associated with ecologically important reef sponges.

Metagenomic analysis of purified viral fractions (metaviromics) recently provided the first insights into the composition and function of viruses inhabiting reef sponges (Laffy et al., 2016; Laffy et al., 2018). Consistent with the pattern reported for sponge-associated microbial communities, the viral communities were found to be highly conserved within each sponge species, and displayed functional repertoires clearly distinct from viruses inhabiting the surrounding seawater (Laffy et al., 2018). Sequence analysis revealed that the metavirome assignments were dominated by viromes from the order Caudovirales but also contained representatives of the Mimiviridae, Phycodnaviridae, Circoviridae, Parvoviridae, Bidnaviridae and Microviridae. Unique viral adaptations to specific host microenvironments were also evident, with viral auxiliary genes being differentially represented across sponge species (Laffy et al., 2018).

While molecular approaches have substantially improved our understanding of viral-host interactions (Breitbart et al., 2002; Rosario & Breitbart, 2011; Laffy et al., 2016), biases associated with DNA/RNA extraction methods (Wood-Charlson et al., 2015) and the limited genomic resources available for most environmental viruses (Roux et al., 2015) can still constrain our understanding of host-associated viral ecology. Transmission electron microscopy (TEM) is a powerful approach that has helped to reveal the morphology and distribution of virus-like particles (VLPs) in many marine hosts as well as deciphering patterns of host-viral interactions (Wilson & Chapman, 2001; Patten, Harrison & Mitchell, 2008; Brum, Schenck & Sullivan, 2013; Pollock et al., 2014; Weynberg et al., 2017b). Here we use TEM to provide the first morphological characterization of viruses associated with 15 different coral reef sponge species and confirm the spatial localization of these VLPs within the sponge holobiont.

Materials and Methods

Sponge collection and identification

Sampling was conducted on coral reefs of Orpheus Island, Great Barrier Reef, Australia (18°35′34″S, 146°28′53″E) and Al Fahal, Red Sea, Saudi Arabia (22°13′95″N, 39°01′81″E), between December 2015 and February 2016. Sampling in Australia was conducted under the Great Barrier Reef Marine Park Authority permit G12/35236.1, and sampling in Saudi Arabia was authorized by the Saudi Arabian coastguard as the study did not involve endangered or protected species.

Triplicate specimens of 15 sponge species were collected by scuba diving between three and 15 m depth. Two sponge species, Stylissa carteri and Carteriospongia foliascens were found at both locations, and sampling was performed in triplicate at both sites. Sponge specimens were photographed in situ before being individually placed within sterile Falcon® tubes and kept on ice until processing. All sampling materials were sterilized prior to and between each sampling. Morphological characterization of sponge species was performed as described in (Hooper & Van Soest, 2002) and DNA barcoding was additionally performed using mitochondrial cytochrome oxidase I (COI) gene primers and internal transcriber spacer 2 (ITS2) region of nuclear ribosomal DNA as described in (Erwin & Thacker, 2007; Andreakis, Luter & Webster, 2012; Wörheide et al., 2012). Sponge species are described in Table 1 and can be seen in Fig. S1.

Table 1 Collection details for all sponge species examined by TEM.

GBR refers to the Great Barrier Reef collection site and RS refers to the Red Sea collection site.

Sponge species	Location	Depth (m)	
Carteriospongia foliascens, P.S. Pallas (1766)	GBR, RS	3–10	
Stylissa carteri, A. Dendi (1889)	GBR, RS	10–15	
Xestospongia sp.	GBR	5–15	
Lamellodysidea herbacea, C. Keller (1889)	GBR	5–10	
Cymbastela marshae, J.N.A.Hooper & P.R. Bergquist (1992)	GBR	10–15	
Cinachyrella schulzei, C. Keller (1891)	GBR	3–7	
Pipestela candelabra, B. Alvarez et al. (2008)	GBR	7–15	
Echinochalina isaaci, J.N.A. Hooper (1996)	GBR	7–15	
Xestospongia testudinaria, J.B.P. Lamarck (1815)	RS	7–15	
Amphimedon ochracea, C. Keller (1889)	RS	7–15	
Hyrtios erectus, C. Keller (1889)	RS	5–15	
Crella (Grayela) cyathophora, H.J. Carter (1869)	RS	7–15	
Mycale sp.	RS	5–15	

Three different sample preparation methods for TEM imaging of sponge-associated viruses were trialed: (i) ultrathin sectioning of sponge tissue (Cheville & Stasko, 2014); (ii) purification of viral fractions via density gradient ultracentrifugation (Lawrence & Steward, 2010; Weynberg et al., 2014) and (iii) filtration of sponge mucus. All samples were examined using a Titan Cubed TEM and images were analyzed on the Cs-corrected Titan™ 80–300 platform at the Imaging Characterization Core Lab in KAUST. TEM search time was standardized to 1 hr/sample.

Preparation of ultrathin sections of sponge tissue

Histological sections were prepared from fresh sponge tissue based on standard procedures for TEM (Cheville & Stasko, 2014). Briefly, each fragment of approximately 1 mm3 was fixed in 2.5% glutaraldehyde in 0.2M cacodylate buffer and kept at 4 °C for 2–24 h. After fixation, samples were immersed in 1% osmium tetroxide in 100 mM phosphate buffer for 1–2 h, washed in distilled water and stained in the dark with 2% aqueous uranyl acetate for 2 h at 4 °C. Stained tissue was dehydrated through a series of ethanol and propylene oxide then embedded in epoxy resin. Ectosome-choanosome oriented sections (about 65 nm thick) were prepared using a Leica EM UC7 ultramicrotome and placed on TEM copper grids.

Viral purification via density gradient solution

Viral purification was performed according to the fraction separation method by sedimentation in density gradients (Meselson, Stahl & Vinograd, 1957) following the pre-processing approach established to isolate viruses from coral and sponge tissue (Weynberg et al., 2014; Laffy et al., 2018). In order to eliminate contaminants present in the aquiferous system, sponges were partially dried via repeated gentle squeezing alternated with rinses of filtered (0.02 µm) seawater. Sponge tissue was then dissected into small pieces (∼5 mm3) and covered with 15 µL of 0.02 µm filter-sterilized (Anotop, Whatman) SM buffer (100 mM NaCl, 8 mM MgSO4, 50 mM Tris pH 7.5), then homogenized with a Craig’s HS30E homogenizer (Witeg, Germany) for 5 to 10 min (min). Tissue homogenate was filtered through a Falcon® 100 µm Cell Strainer (Corning, USA), then centrifuged at 500 g for 15 min at 4 °C to pellet the majority of cell debris. The supernatant was used to purify the VLP via centrifugation in Cesium Chloride solution, with density varying from 1.2 g/mL to 1.6 g/mL (Weynberg et al., 2014). After ultracentrifugation, sponge VLPs were collected from the fractions with densities between 1.2 g/mL and 1.5 g/mL. In order to exchange the buffer and remove CsCl salts, samples were loaded onto 30 KDa Amicon centrifugal spin columns (Millipore, EUA) and centrifuged at 4,000 g for 30 min at 4 °C. This process was repeated four–six times per sample. Filter- sterilized SM Buffer was added to the concentrate and all flow-through was discarded. The concentrate was fixed in 0.5% glutaraldehyde and kept at 4 °C until TEM analysis. TEM preparation involved applying a droplet of sample onto a TEM Copper grid, rinsing with sterile water, staining with 1% uranyl acetate for one min, washing with sterile water, followed by removal of excess liquid from the grid by touching filter paper to the edge.

Viral purification via filtration of sponge mucus

To describe the VLPs associated with sponge mucus and the external ectoderm, the sponge surface was carefully scraped with a sterile scalpel blade followed by rinsing three times with filtered (0.02 µm) seawater. This TEM preparation method was based on a viral purification method described for marine hydras (Grasis et al., 2014). Extracted mucus was added to filtered (0.02 µm) Milli-Q® water (1:4) and centrifuged at 4,000 g for 10 min. Mucus supernatant was filtered through 0.45 µm filters (EMD Millipore, Burlington, CA, USA) and fixed in 1.5% glutaraldehyde. TEM imaging of mucus preparations was performed as described above for CsCl purified samples.

Results

Sponge associated viruses

TEM analysis revealed that viral particles are diverse constituents of the sponge holobiont. Fifty VLP morphotypes (Figs. 1–5; Table S1; Morphotypes: M-I–M-L) were found in association with eight coral reef sponge species from the Great Barrier Reef: Carteriospongia foliascens, Stylissa carteri, Xestospongia sp., Pipestela candelabra, Lamellodysidea herbacea, Cymbastella marshae, Echinochalina isaaci and Cinachyrella schulzei; and seven sponge species from the Red Sea: Carteriospongia foliascens, Stylissa carteri, Xestospongia testudinaria, Hyrtios erectus, Mycale sp., Amphimedon ochracea and Crella cyathophora. VLPs were observed within sponge cells, in the extracellular mesohyl matrix, in the mucus/surface biofilm and within sponge-associated microbes. A diverse range of viral morphologies were observed, including hexagonal (tailed and non-tailed), spherical, filamentous, brick-shaped, beaded and geminate VLPs. While we detected numerous viral morphotypes, most were rare and often obscured by vesicles, cell debris and particulate organic matter.

Figure 1 Representative morphotypes of virus-like particles associated with GBR and Red Sea sponges.

GBR sponge species: (A, J, K, L) C. foliascens, (B) Xestospongia sp., (C, F, G) E. isaaci, (E) C. schulzei. Red Sea sponge species: (D, H) S. carteri, (K) Amphimedon ochracea. TEM preparation method: (A, J, K) ultrathin sections of sponge tissue, (B–I, L) viral purification via filtration of sponge mucus. Scale bar: 200 nm. Black arrows indicate the viral tail and white arrows indicate the VLPs.

Figure 2 Representative morphotypes of virus-like particles associated with GBR sponges.

Sponge species: (A) C. foliascens, (B, C) Stylissa carteri, (D) Xestospongia sp., (E–H) Pipestela candelabra, (I–K) Lamellodysidea herbacea, (L) C. schulzei. TEM preparation method: (A, H–K) viral purification via filtration of sponge mucus, (B–F, L) viral purification via CsCl gradient centrifugation, (G) ultrathin sections of sponge tissue. Scale bar: 200 nm.

Figure 3 Representative morphotypes of virus-like particles associated with GBR and Red Sea sponges.

GBR sponge species: (A, B) C. schulzei, (C) Cymbastella marshae. Red Sea sponge species: (D, E) C. foliascens, (F–H) S. carteri, (I) Xestospongia testudinaria, (J–L) Hyrtios erectus. TEM preparation method: (A, D–E, I–L) ultrathin sections of sponge tissue, (B, C, F–H) viral purification via filtration of sponge mucus. Scale bar: (A–C, E–L) 200 nm, (D) 500 nm. Black arrows indicate the VLPs.

Figure 4 Representative morphotypes of virus-like particles associated with GBR and Red Sea sponges.

GBR sponge species: (A) Mycale sp., (B) C. foliascens, (C, D) Xestospongia sp., (E) C. schulzei. Red Sea sponge species: (F, G) C. foliascens, (H, I) S. carteri, (J–L) Xestospongia testudinaria. TEM preparation method: (A, B, D, E, H, I) viral purification via filtration of sponge mucus, (C) viral purification via CsCl gradient centrifugation, (F, G, J–L) ultrathin sections of sponge tissue. Scale bar: (A–E, H, I) 200 nm, (F, G, J–L) 500 nm.

Figure 5 Representative morphotypes of virus-like particles associated with GBR and Red Sea sponges.

GBR sponge species: (C) Lamellodysidea herbacea, (I) C. foliascens. Red Sea sponge species: (A, B, H) Crella cyathophora, (D–G) Amphimedon ochracea, (J–L) Hyrtios erectus. TEM preparation method: (A, C, D) viral purification via filtration of sponge mucus, (B, E–L) ultrathin sections of sponge tissue. Scale bar: (D) 100 nm, (A–C, E, H–L) 200 nm, (F, G) 5 µm. ECM: External Cell Matrix, om: outer membrane, im: inner membrane, cm: core membrane, lb: lateral bodies; c: core, e: external membrane; b: bacterium. Black arrows indicate the VLPs.

Most sponge-associated VLP morphotypes possessed an icosahedral/polyhedral symmetry (∼75%), ranging from 60–205 nm in diameter (Figs. 1– 3, 4A). Tails were evident on some VLPs, confirming the presence of viruses from the bacteriophage order Caudovirales. Tailed VLPs were tentatively assigned to the three Caudovirales families based on their capsid symmetry and tail size/shape. VLPs characteristic of the Podoviridae presented a short tail attached to a non-enveloped icosahedral capsid and these VLPs were observed in the sponges C. foliascens (Fig. 1A), Xestospongia sp. (Fig. 1B), E. isaaci (Fig. 1C) and S. carteri (Fig. 1D). VLPs characteristic of the Siphoviridae presented an icosahedral head with a long non-contractile tail and these VLPs were detected in the surface biofilm of C. schulzei (Fig. 1E). VLPs characteristic of the Myoviridae presented an icosahedral head and a long contractile tail and these VLPs were observed in the sponges E. isaaci (Figs.  1F, 1G), S. carteri (Fig. 1H) and A. ochracea (Fig. 1I).

Non-tailed icosahedral/polyhedral VLPs were observed using all three TEM preparation methods. Particle sizes ranged from 60 to 205 nm in diameter and some presented an electron dense core inside the viral capsid (35–124 nm in diameter). The majority of VLPs did not show an envelope outside the capsid, however an envelope was observed in association with a small proportion of VLPs. (Figs. 2E, 3K–3L, 5H). A typical example of an enveloped VLP was observed in Hyrtios erectus where a group of four virions were observed within a vacuole in the mesohyl matrix (Fig. 3L) and another free virion was captured merging its envelope into the cell membrane of the host (Fig. 3K).

In addition to the polyhedral VLPs, eight morphotypes of filamentous virus-like particles (FVLPs) were observed in the sponge mucus, mesohyl matrix, within sponge cells and associated with sponge-associated microorganisms (Figs. 4C–4L; 5A). These morphotypes varied greatly in size (100–1300 nm length, 12–60 nm width) and shape. Rod-shaped FVLPs were detected in the CsCl purified viral fraction of Xestospongia sp. (Figs. 4C, 4D) and the mucus of S. carteri (Fig. 4I). Although similar, the S carteri bacilliform VLPs were longer than those observed in Xestospongia sp. (230 nm long, 19 nm wide in S, carteri; 120–130 nm long, 18 nm wide in Xestospongia sp.). In C. foliascens, a FVLP was frequently observed attached to cyanobacteria and within the sponge mesohyl, (Figs. 4F–4G). This FVLP resembled viruses of the family Inoviridae due to their shortened body (100–130 nm length, 50–60 nm width) and electron-translucent core with outer membrane structures consistent with a glycoprotein coat surrounding the entire membrane (Ploss & Kuhn, 2010). In X. testudinaria, a FVLP morphotype was observed within cells and dispersed throughout the mesohyl (Figs. 4J–4L). This thin, elongated FVLP (340–1,300 nm long and 15–30 nm wide) was observed at high abundance inside some choanocyte cells and lysed cells releasing virions were also evident (Figs. 4J–4L). Another distinct FVLP morphotype was evident in the sponge mucus of C. cyathophora (Fig. 5A). It presented a tube-like shape indicating helical symmetry and size ranging from 150–154 nm in length and 22–25 nm in width.

Geminate VLPs were observed in C. cyathophora mesohyl matrix (Fig. 5B), in L. herbacea mucus (Fig. 5C), and found infecting filamentous cyanobacteria associated with the sponge A. ochracea (Figs. 5D–5G). The cyanobacteria associated VLPs shared morphological traits with viruses from the family Geminiviridae (Li, Ou & Zhang, 2013) and were typically twinned (81–95 nm long, 37–48 nm wide), comprising two quasi-isometric particles (34–45 nm length). The VLPs were spread across the cytoplasm, thylakoid lumen, and vacuoles of the cyanobacterial cells and were often at high abundance surrounding the stellar bodies (Fig. 5F).

A brick-shaped VLP morphotype, closely resembling viruses from the Poxviridae, was observed in sections of Crella cyathophora (Fig. 5H). This morphotype had a complex structure comprising a biconcave core encased within a double layer membrane with two lateral bodies surrounded by an ovoid envelope ( Buller & Palumbo, 1991). Three representatives of this morphotype were observed within the sponge mesohyl matrix, and a single non-enveloped VLP was also observed in close proximity to a lysed sponge cell.

A beaded VLP was observed in sections of the sponges C.  foliascens (Fig. 5I) and H. erectus (Figs. 5J–5L). In C. foliascens, the branched VLP was 340 nm long, and was comprised of six beads, each measuring 30–35 nm in diameter. In H. erectus, the VLPs varied from 80 to 350 nm in length and were composed of 2–8 aligned beads with diameters ranging from 36–42 nm. This morphotype was observed as isolated VLPs, attached to extracellular vacuole membranes in the sponge mesohyl, and within intracellular vacuoles of archaeocyte cells.

Discussion

Sponges are complex holobionts that host a diverse array of bacteria, archaea, and eukaryotic microorganisms (Fan et al., 2012; Fan et al., 2013; Webster & Thomas, 2016). Whilst previous publications have alluded to the potential importance of viruses in sponges (Claverie et al., 2009; Webster & Taylor, 2012; Laffy et al., 2016; Laffy et al., 2018), including in sponge disease (Luter, Whalan & Webster, 2010), this study provides the first visual evidence that viruses are diverse components of the sponge holobiont. The broad range of VLP morphologies visualised across the 15 different sponge species is consistent with recent molecular data showing sponges harbour diverse communities of viruses (Laffy et al., 2016; Laffy et al., 2018).

The frequent detection of multiple viral morphotypes within a single sponge species most likely reflects the large number of potential hosts within the sponge holobiont (sponge cells, bacteria, archaea, microeukaryotes). However, it is also possible that multiple viruses infect the same host, as has been observed in some bacterioplankton (Holmfeldt et al., 2007) and corals (Thurber & Correa, 2011). Similarly, the same viral morphotype may infect multiple hosts within the holobiont, as recently highlighted from phage-bacteria network analyses (Flores et al., 2011; Flores, Valverde & Weitz, 2013). This is particularly relevant considering the role of viruses in lateral gene transfer between hosts and their subsequent effects on host metabolism (Breitbart et al., 2018). Observed viral morphotypes may also not be native to the holobiont, as some may have been extracted from the virioplankton by the sponge’s aquiferous system. Although the isolation methods employed in this study unveiled a wide range of VLP morphotypes, no quantitative assessments were undertaken. To further our understanding of viral dynamics within the sponge holobiont, quantitative studies that count the number of VLPs per known tissue area, perform quantitative transmission electron microscopy (qTEM) (Brum, Schenck & Sullivan, 2013), flow cytometry (Brussaard, 2004; Pollock et al., 2014) or fluorescent staining (Leruste, Bouvier & Bettarel, 2012; Pollard, 2012) should also be performed.

Morphology is an important feature for viral classification according the International Committee on Taxonomy of Viruses (ICTV). However, there are also some limitations associated with using TEM to identify viruses. For instance, many viral groups lack morphological structures that characterize them as typical viral particles by TEM. Also, as many viruses are small and simple they can be mistaken for non-viral particles such as cellular vesicles or organelles. Although the assignment of viral-like particles in this study was made by comparison to morphologically characterised viruses, the possibility remains that some VLPs may not represent true viruses.

In this study, TEM analysis revealed a prevalence of polyhedral VLPs with characteristic bacteriophage morphology, consistent with what has been described for other marine invertebrates (Wilson et al., 2005; Davy et al., 2006; Davy & Patten, 2007; Patten, Harrison & Mitchell, 2008). The presence of Caudovirales-like morphotypes highlights the potential for these VLPs to target sponge symbionts and ultimately control microbial population dynamics within the sponge holobiont. Amongst them, a Siphoviridae”- VLP detected in the surface biofilm of C. schulzei presented similar morphology, although slightly smaller, to the previously described sponge-associated Phage ΦJL001 ( Lohr, Chen & Hill, 2005).

Surprisingly, relatively few tailed bacteriophage were detected within the reef sponges, despite the dominance of Caudovirales within the assigned sponge viromes (Laffy et al., 2018). Although the dominance of tailed viruses in aquatic ecosystems is well characterised (Mizuno et al., 2013; Weynberg et al., 2017a; Weynberg et al., 2017b; Thurber et al., 2017; Laffy et al., 2018), results from morphological analysis of uncultivated viruses vary with respect to the relative dominance of tailed ( Cochlan et al., 1993; Colombet et al., 2006; Dutova & Drucker, 2013) versus non-tailed (Bergh et al., 1989; Wommack et al., 1992; Auguet, Montanié & Lebaron, 2006; Brum, Schenck & Sullivan, 2013) VLPs. The reduced number of tailed VLPs in morphological descriptions has been attributed to the destruction of the delicate VLP structures during centrifugation and TEM sample preparation (Cochlan et al., 1993; Proctor, 1997). However, Brum, Schenck & Sullivan (2013) have shown that sample preservation and preparation do not alter the morphological characteristics of seawater derived VLPs (Brum, Schenck & Sullivan, 2013) and non-tailed VLP have therefore been proposed as the dominant viral group in aquatic ecosystems (Brum, Schenck & Sullivan, 2013; Kauffman et al., 2018). Nevertheless, in this study, tailed VLPs were almost exclusively detected in samples purified via filtration of mucus or scraping of the external biofilm, the least disruptive of the three TEM preparation methods. This suggests that tailed VLPs are either more abundant on the external surface of the sponge or that the TEM preparation method could bias the detection of tailed VLPs in sponges by mechanically damaging or distorting viral structures.

Filamentous viral-like particles (FVLP) were detected in both prokaryotic and eukaryotic cells within the sponge holobiont. In C. foliascens, multiple individual Inoviridae-like VLPs were observed attached to the surface of cyanobacteria, although no virions were observed inside the cells. The absence of intracellular FVLPs combined with the absence of a dense core in these morphotypes provides further support for their classification as putative Inoviridae, as the replication mechanism of this viral family often relies on the virus injecting its DNA into the host cell and getting extruded without inducing cell lysis (Bayer & Bayer, 1986; Russel, 1991; Ploss & Kuhn, 2010). A previous study demonstrated that temperate viruses are relatively less abundant within host cells at high density (McDaniel et al., 2002).

FVLPs with helicoidal symmetry resembling Spiraviridae were detected in the sponge C. cyathophora, with this viral family known to infect Archaea (Mochizuki et al., 2012). FVLPs were also observed infecting eukaryotic cells in X. testudinaria. Abundant elongated and flexible FVLPs were also detected in the archaeocytes and extracellular mesohyl matrix of X. testudinaria (Figs. 4J–4L). The point of host cell lysis was captured with a recently burst cell releasing virions into the extracellular matrix (Figs. 4J–4K), characteristic of typical lytic viral infection (Dyson et al., 2015). Morphologically similar filamentous VLPs have been detected in coral mucus and associated Symbiodinium and were characterised as a coral-infecting RNA virus (Davy et al., 2006; Weynberg et al., 2017b). There is a general lack of studies investigating filamentous viruses in marine invertebrates, although metaviromic sequencing recently detected sequences assigned as filamentous viruses of the family Inoviridae in Great Barrier Reef sponges (Laffy et al., 2018).

VLPs morphologically consistent with viruses from the family Geminiviridae were observed in association with cyanobacteria in the sponge A. ochracea. Geminiviridae-like viruses have been isolated from infected freshwater cyanobacteria (Li, Ou & Zhang, 2013), and, with the exception of being slightly smaller (79 ±5 nm in length, 28 ±3 nm in diameter), the geminate VLPs from A. ochracea were morphologically similar. Most infected cyanobacterial cells had dense populations of these VLPs (Figs. 5D–5G), although no lysed cells or free geminate VLPs were observed in the sponge mesohyl. However, several extracellular vesicles containing VLPs were observed, indicating that VLPs could use cell extrusion as part of their reproductive cycle. A geminate VLP has previously been isolated from mucus secreted by scleractinian corals (Davy & Patten, 2007), however the morphology differs from the A. ochracea VLP, since it is notably bigger (about 145 nm in length, 82 nm width), with each isomer being wider than they are long, contrasting with the isomer dimensions in the A. ochracea VLP. Beaded VLPs were also detected in sponges and their non-isomeric particles comprising a flexible filament strongly resembled the beaded VLPs previously reported from scleractinian corals (Davy & Patten, 2007; Lawrence et al., 2015).

Brick-shaped VLPs closely resembling viral morphotypes from the family Poxviridae were observed within the mesohyl of Crella cyathophora, a (Fig. 5H). Typical of enveloped viruses, poxviruses use their envelopes to connect and fuse with their host membrane so that the viral capsid is injected directly into the host cell (Moss, 2012). Poxviruses are notable pathogens, infecting a wide host range among vertebrate and invertebrate taxa (Bracht et al., 2006; Grasis et al., 2014; Haller et al., 2014). In the marine environment, they have been reported associated with cetaceans and pinnipeds (Bracht et al., 2006) and more recently, analysis of sponge metaviromes detected sequences affiliated to Poxviridae in Amphimedon queenslandica and Ianthella basta (Laffy et al., 2018).

Conclusion

In this study we validated the efficacy of three different methods for TEM imaging of sponge-associated viruses: (i) ultrathin sections of sponge tissue, (ii) purification via density gradient ultracentrifugation and (iii) ectoderm scraping and filtration of sponge mucus. While density gradient purification facilitated concentration and recovery of VLPs from different areas of the sponge holobiont, it also co-concentrated cellular debris, potentially masking many VLPs. Tissue sectioning enabled direct visualisation of spatial localisation and host-viral interactions but was labour intensive and some VLP structures were distorted during sectioning. Ectoderm scraping and collection of sponge mucus was most effective at preserving delicate viral structures and minimizing the amount of cellular debris, however, it was restricted to recovering VLPs associated with the sponge mucus or ectoderm.

This first morphological characterisation of sponge-associated viruses revealed a wide diversity of VLPs infecting both the sponge cells and symbiont compartments of the holobiont. By confirming that viruses are a significant component of the sponge holobiont, this work paves the way for future metaviromic and cell culturing analyses that can characterise the taxonomy and function of the sponge viral community.

Supplemental Information

Figure S1 Sponge species where the VLPs were investigated

Red Sea sponge species: (A) Amphimedon ochracea, (B) Xestospongia testudinaria, (C) Crella cyathophora, (D) Hyrtios erectus, (E) Mycale sp. GBR and Red Sea sponge species: (F) Stylissa carteri, (G) Carteriospongia foliascens. GBR sponge species: (H) Echinochalina isaaci, (I) Cymbastella marshae, (J), Cinachyrella schulzei, (K), Lamellodysidea herbacea, (L), Pipestela candelabra, (M) Xestospongia sp.. Scale bar = 10 cm. Photos by Cecília Pascelli.

Click here for additional data file.

Table S1 Morphological characterization of virus-like particles associated with coral reef sponges from the Great Barrier Reef (GBR) and the Red Sea (RS)

Click here for additional data file.

The authors wish to acknowledge Dr. Karen Weynberg for support with the microscopy sample preparation and Dr. Rachid Sougrat and Dr. Ptissam Bergam for support with the operation of the TEM.

Additional Information and Declarations

Competing Interests

Author Contributions

Field Study Permissions

Data Availability

The authors declare there are no competing interests.

Cecília Pascelli conceived and designed the experiments, performed the experiments, analyzed the data, contributed reagents/materials/analysis tools, prepared figures and/or tables, authored or reviewed drafts of the paper, approved the final draft.

Patrick W. Laffy analyzed the data, authored or reviewed drafts of the paper, approved the final draft.

Marija Kupresanin performed the experiments, authored or reviewed drafts of the paper, approved the final draft.

Timothy Ravasi contributed reagents/materials/analysis tools, authored or reviewed drafts of the paper, approved the final draft.

Nicole S. Webster conceived and designed the experiments, analyzed the data, contributed reagents/materials/analysis tools, authored or reviewed drafts of the paper, approved the final draft.

The following information was supplied relating to field study approvals (i.e., approving body and any reference numbers):

Sampling in Australia was conducted under the Great Barrier Reef Marine Park Authority permit G12/35236.1

The following information was supplied regarding data availability:

The raw data are provided in Table S1.

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
