# Peer review of "Morphological characterization of virus-like particles in coral reef sponges"

_PeerJ, doi:10.7717/peerj.5625_

## Round 0.1 · original submission · Minor Revisions

All three reviewers found this to be an interesting and valuable contribution, yet suggested changes that could improve the readibility and organization of the manuscript. Please revise accordingly.

Reviewer 1 ·

Basic reporting

no comment

Experimental design

no comment

Validity of the findings

no comment

Additional comments

The manuscript by Pascelli et al describes virus-like particles observed within coral reef sponges. This is clearly an initial, observational study of these viruses, but does provide a first look at the possible viral morphotypes inhabiting the sponge holobiont. I have a few comments regarding issues I think need to be addressed within the manuscript, mostly to increase clarity regarding what the results represent.

General Comments:

1) In the Abstract and Introduction especially, the use of “host” is a bit confusing at times regarding whether it means “species of sponge where the virus is found” versus “organism that the virus infects”. Can the authors find a way to standardize this language throughout the manuscript to avoid confusion? For example, in Line 76, the use of the term “host-specific” to refer to the unique community of viruses within sponges is misleading (or incorrect). To determine if a virus is “host-specific” there needs to be evidence showing that the virus only infects one host. I know that the title of the Laffy et al paper uses that term, but they are correct later in the paper when they use the terms “habitat-specific” and “host species specific”.

2) I think there should be some statement in the Discussion regarding the possibility that some of these VLPs might not be viruses. Yes, that is inclusive in the definition of “VLP”, but this nuance may be lost on non-specialist readers, and thus an explanation is in order. For example, Figure 3K shows an obvious virus (and it is a very nice image!), but Figure 2D could be some type of inclusion bodies. This is a limitation of using TEM to identify viruses; obvious head-tail structures are easily identified, but it is difficult to distinguish other morphologies because there are many other things in cells and in the environment that are within the same size range of viruses and have dark staining in TEM images.

3) Within the Abstract, the phrases “varied in prevalence” (lines 27-28) and “abundant” (line 34) are a bit misleading and should be changed/removed because the authors do not use quantitative methods to investigate the relative or absolute abundances of viral morphotypes in the samples. I appreciate the authors not describing their results as quantitative throughout the remaining part of the manuscript, but I also think that the authors can insert a statement in the Discussion regarding (i) the fact that the dataset is observational and (ii) suggesting that future studies be conducted in a quantitative manner (i.e., analyzing a known area for each thin section sample, using a known/consistent volume from the mucus and aquiferous samples, not using CsCl gradients because this can remove specific virus types that fall outside of the density range selected such as lipid-containing viruses, using ultracentrifugation to deposit viruses onto grids instead of adsorption, and observing a fixed number of viruses per sample such that the percent of each morphotype can be determined – see Brum et al 2013, ISME J, for example).


Minor Comments:
Line 78: The paper by Laffy et al shows that most of the *identifiable* virus sequences were Caudovirales. Like most environments, most of the viral sequences in that study were taxonomically “unknown”.

Line 139: Please describe the method for sample preparation for “TEM analysis”. I assume it is the adsorption method described in the next section (lines 147-148).

Lines 147-148: Please add a few more details regarding the TEM grid preparation. What method was used to render the grids hydrophilic? How long was the mucus sample allowed to adsorb to the grid? Was the grid then rinsed a few times with purified water? I also assume that the grid was rinsed after staining with UA.

Line 214: A beaded virus that varies in the number of beads would be a completely new discovery (as far as I know). There should be some discussion of this, regarding whether this is a new virus family, or possibly a part of the sponge itself. For example, these look a bit like chains of magnetosomes (e.g., Lohsse et al, 2011, PLoS One).

Lines 235-240: The significant loss of virus tails through deposition of viruses onto grids via ultracentrifugation was shown *not* to occur in seawater samples (Brum et al, 2013, ISME J). And in fact the non-tailed viruses dominated the viral communities throughout the oceans. Recently, a new family of non-tailed bacteriophages has been discovered (Kauffman et al, 2018, Nature), further supporting the hypothesis that many marine viruses may be non-tailed bacteriophages. The authors may wish to amend this section of the manuscript to state that their specific preparation methods (i.e., CsCl, centrifugal concentration devices, etc.) may have resulted in damage to the viruses, as this has not yet been tested.

Lines 288-289: Most bacteriophages studied have a wide variety of host ranges (see Flores et al, 2011, PNAS, for a metaanalysis of this).

Figure 2B: It is a bit difficult to distinguish the arrow heads in this image. At first I thought they were part of the virus. Can the authors make these larger, or make them stand out from the image in some way?

·

Basic reporting

This is a really nice piece of work, conducted in a very conscientious way, and which provides novel information about the diversity of sponge-associated viruses.
One could regret the absence of viromic data to accompany the exhaustive morphological description of viruses; however this study is still justified on its own and has sufficient substance for publication in Peer J.
By collecting 15 different sponge species from two constrasted sites (the Great barrier in Australia and the Red Sea in Saudi Arabia), and by using three different protocols for TEM observation, authors have accomplished remarkable sampling and methodological efforts to characterize the viral morphotypes. Micrographs are also of very good quality which renders the conclusions very convincing. I also compliment the methodological precautions to get ridd off the surrounding viruses of the water column, present in the filtering system of the sponge, and which could be a potential source of bias. I also appreciated the conclusion about the accuracy of the three tested protocols which will be very helpfull for further morphological studies of viruses in reef invertebrates.
A interesting paper related to this topic (ie, phages in sponge microbial diversity) is missing : Lohr et al 2005, Genomic analysis of bacteriophage JL001: Insights into its interaction with a sponge-associated Alpha-Proteobacterium, Appl. Environ Microbiol 71 : 1598–1609.
Finally I don’t have much to criticize, this was a very pleasant paper to read.

Experimental design

no comment

Validity of the findings

no comment

Reviewer 3 ·

Basic reporting

.

Experimental design

.

Validity of the findings

.

Additional comments

Overview. This manuscript describes the results of a microscopy based study that evaluated the viral like particles associated with several species of sponge. Descriptive in nature, this work uses the images to hypothesize what kinds of phage and eukaryotic viruses are associated with these basal metazoans and their symbionts. Overall the work is conducted admirably and the work is novel. Despite this the work requires some editing. While I am enthusiastic about its publication, I feel that the manner in which the authors present the data is quite difficult to read, even for an expert. The authors present the collection of images (figures 1-5) in terms of which sponge species they are evaluting. However, the text refers to the images in terms of hypotheses about what kind of viruses they believe these images represent, for example myophage. I think the authors should instead present first an overall figure showing images from many of the sponges and then the subsequent figures should focus the various kinds of vlps in all of the different sponges. This would align with the presentation of the data in the results text more completely. It would also make it much easier for the readers to follow. An example of this confusing format of the paper comes from lines 165-166 where the authors want the reviews to be able to find 60-200nm vlps in all the figures “Fig. 1A-C, E, G, J; 2A-L; 3B-F, I-L; 4A-C, F, J-166 L; 5D, I, K” by flipping between them. This is very frustrating and time consuming when the authors could have just put those together to make their point more clearly. Any reference to figures that contains more than 1 figure or more than one subsection of multiple figures is simply too hard to follow. Thus, my primary recommendation is that the authors remake all their figures to more conclusively support their hypotheses about to which categories/families these vlps belong.
Specifics:
Line 114 and 133. These are strange references to density gradient ultracentrifugation which has been around for decades. Are there differences in the gradients that warrant these references? What densities were used? What density where the vlps isolated from?
Lines 122-123. Was there no post fixation process conducted to remove the spicules? This seems surprising.
Line 31, 162 and 209. Dumbo-eared? Is this an official term? Many people that are not from western cultures many not get this reference. Elephant eared?
Line 260. Replace Zooxanthellae with Symbiodinium.
Figures. Again, I think these all need to be reorganized. Also, these roman numeral indicators are really not easy to follow. There must be a less complicated way of doing this to make everything easier for the reader to follow.

---

## Round 0.2 · accepted · Accept

Thank you for your careful attention in addressing the reviewers' concerns.

#